# Chemical Composition and Antioxidant and Antibacterial Potencies of the *Artemisia ordosica* Aerial Parts Essential Oil during the Vegetative Period

**DOI:** 10.3390/molecules27248898

**Published:** 2022-12-14

**Authors:** Jize Zhang, Qiang Pan, Xiaoqing Zhang

**Affiliations:** Institute of Grassland Research, Chinese Academy of Agricultural Sciences, Hohhot 010010, China

**Keywords:** *Artemisia ordosica*, essential oil, GC-MS analysis, antioxidant, antibacterial activity

## Abstract

As one of the vital shrubs growing in crusted areas in China, *Artemisia ordosica* (belonging to the Asteraceae family) is abundant in essential oil, and its aerial part’s essential oil has been reported to have some biological activities during the flowering and fruit set stage, and has been used in folk medicine. However, little is known about the biological activities of its aerial part’s essential oil during the vegetative period. Thus, the purpose of this work was to determine the chemical composition and evaluate the antioxidant and antibacterial potencies of the essential oil extracted from *A. ordosica* aerial parts during the vegetative stage. Gas chromatography coupled with mass spectrometry (GC-MS) revealed that spathulenol (9.93%) and *α*-curcumene (9.24%), both sesquiterpenes, were the most abundant of the 74 chemical constituents detected in the essential oil of *A. ordosica*. The antioxidant activity of the essential oil was found to be relatively moderate against 2,2-diphenylhydrazyl (DPPH), 2,2′-azino-bis(3-ethylbenzothiazoline-6-sulfonic acid) (ABTS), and hydroxyl radical (OH^●^) radicals. The essential oil exhibited strong antibacterial activity against *Staphylococcus aureus, Salmonella abony* and *Escherichia coli*, with minimum inhibitory concentrations (MICs) of 2.5, 5, and 10 μL/mL, respectively. The results indicate that the essential oil of *A. ordosica* possesses notable antibacterial properties as well as antioxidant capability and can thus be employed as a natural ingredient which can be used as a substitute for antibiotics in the animal feed industry. However, in vivo toxicological studies are still required to determine the safety level and beneficial outcomes of the *A. ordosica* essential oil for future utilization.

## 1. Introduction

The genus *Artemisia* is a division of the Asteraceae family, which has a huge variety of different plant species, including over 380 species around the world [1]. Numerous species are abundant in secondary metabolites with diverse modes of action, including *Artemisia annua* L. (well-known as “qinghao”), *Artemisia arborescens* L., and *Artemisia biennis* Willd., which have vast applications in the medicine, cosmetics, and pharmaceutical industries [2]. *Artemisia ordosica*, an arido-active, essential-oil-rich Asteraceae shrub, is native to the arid and semiarid areas of northern China, with a widespread distribution in Inner Mongolia, Shanxi, Ningxia, and Gansu. In these places, the plant’s aerial parts are widely used as folk medicines to cure a variety of health conditions, such as rheumatism, fever, and swelling [3]. In addition, the *A. ordosica* essential oil displays a significant number of biological properties, including allelochemicals and larvicidal effects, and has tremendous promise as a biological herbicide and pesticide in agriculture [4,5]. Previous studies have shown that the flowering and fruit set-stage essential oil of *A. ordosica* is rich in terpenoids, alcohols, ketones, esters, and carboxylic acids, and the chemical composition varies significantly with various factors, such as the growth environment and harvest season [6].

Variation in abiotic environmental conditions throughout the plant growth stage influences secondary metabolism pathways, resulting in alternations in the amount and bioactivity of bioactive substances in the harvested plant at various phenological stages [7]. Yang et al. [5] and Zhang et al. [4] identified 37 and 26 components, respectively, in *A. ordosica* during the flowering and fruit set stages in the Kubuqi Desert, Inner Mongolia. Previous findings demonstrated that the essential oil of some plants collected during the vegetative period had the optimum antioxidant or antibacterial activities, indicating the better application of plants in the vegetative stage [8,9,10]. Despite the above reports on the allelochemicals and larvicidal effects of *A. ordosica*, the antioxidant and antibacterial activities of the *A. ordosica* essential oil in the vegetative phenophase have been rarely recorded. Moreover, monitoring the phytochemical content and quality during phenological growth stages can indicate the pattern of accumulation of these compounds for the acquisition of specific phytochemical components with the desired quality for the pharmaceutical or animal feed industries. The utilization of antibiotics in farm animals as “growth promoters” has increased microorganism resistance, and it is necessary to search for alternative agents from medicinal plants to control microorganisms. Therefore, the present study aimed to characterize the content and constituents of the essential oil from aerial parts of *A. ordosica* collected during the vegetative stage, as well as determine the oil’s free radical scavenging and antibacterial properties for its potential application to replace antibiotics in farm animals with non-synthetic compounds.

## 2. Results

### 2.1. Yield of Essential Oil and Chemical Composition

The hydrodistillation of the aerial parts of *A. ordosica* collected during the vegetative stage provided a yellow-colored essential oil (yield: ~0.33%, *v*/*w*). Upon GC-MS analysis, the main components and the composition fraction of the *A. ordosica* essential oil were analyzed. A GC-MS chromatogram of the *A. ordosica* essential oil is shown in Appendix A. Seventy-four constituents accounting for 94.53% of the total oil were identified (Table 1). The essential oil of *A. ordosica* was characterized by oxygenated sesquiterpenes (33.95%), followed by sesquiterpene hydrocarbons (17.86%), monoterpene hydrocarbons (16.10%), and oxygenated monoterpenes (11.59%), representing 79.5% of the compounds (Table 1). Among these components, spathulenol (9.93%) and *α*-curcumene (9.24%) were the main components. Other higher content components (more than 4%) included α-bisabolol (6.45%), *cis*-(+)-nerolidol (5.26%), and d-limonene (4.25%). The chemical structure of these prominent constituents in the essential oil is shown in Figure 1. Forty-seven constituents of the oil amounted to <1% (Table 1).

### 2.2. Antioxidant Activity

To further evaluate the potential utilization of the *A. ordosica* essential oil, its antioxidant and radical scavenging activity against 2,2-diphenylhydrazyl (DPPH), 2,2′-azino-bis(3-ethylbenzothiazoline-6-sulfonic acid) (ABTS), and hydroxyl radical (OH^●^) were investigated. The free radical scavenging activities of the *A. ordosica* essential oil are shown in Figure 2, with ascorbic acid being the standard. The essential oil had relatively high DPPH and ABTS free radical scavenging abilities (>70%) (Figure 2A,B). The essential oil at the maximum concentration (12 mg/mL) was most effective against DPPH free radicals at 70%. The most effective activity against ABTS free radicals was 77%. The IC_50_ values of the essential oil for DPPH and ABTS free radicals were approximately 5.9 mg/mL and 5.7 mg/mL, respectively. As shown in Figure 2C, the essential oil exhibited lower OH^●^ radical scavenging activity (63%), with an IC_50_ value of approximately 6.8 mg/mL.

### 2.3. Antibacterial Activity

The antibacterial activity of the *A. ordosica* essential oil was determined against multiple microorganisms. The essential oil was active against all microbial strains. However, the efficiency of the oil was found to be variable to different degrees. The results of the antibacterial activities are shown in Table 2. The minimum inhibitory concentration (MIC) values of the essential oils are summarized in Figure 3. The *A. ordosica* essential oil presents potent inhibition against both Gram-negative and Gram-positive microorganisms. Above 5 μL/mL, the oil effectively inhibited the bacterial growth of *Staphylococcus aureus* and *Salmonella abony*. However, no visible bacterial growth of *Staphylococcus aureus* was observed only at an oil concentration of 2.5 μL/mL, which could be considered as the MIC value of *Staphylococcus aureus*. *Salmonella abony* was less susceptible to the essential oil, with an MIC value of 5 μL/mL. The most resistant strain to the *A. ordosica* essential oil was *Escherichia coli*, with an MIC value of 10 μL/mL.

## 3. Discussion

The chemical characteristics and yield of the *A. ordosica* essential oil obtained in our study (during the vegetative stage) differed from those of prior studies. Zhang et al. [4] reported that the aerial parts of the *A. ordosica* essential oil (collected at the fruit set stage) had a yield of 0.73%, with a total of 26 components identified, including caryophyllene (17.81%), *β*-bisabolene (12.11%), spathulenol (10.56%), *β*-caryophyllene oxide (8.67%), (E)-phytol (5.64%), and *β*-elemene (5.56%) as the major components. Yang et al. [5] discovered 37 components in the essential oil of *A. ordosica* (collected at the flowering stage), of which 2,5-etheno[4.2.2]propella-3,7,9-triene (24.81%), *trans*-nerolidol (10.39%), *α*-longipinene (8.82%), spathulenol (8.17%), *β-trans*-ocimene (5.09%), and D-limonene (4.32%) were the primary components. Farhadi et al. [7] revealed that the number of *Achillea millefolium* (Asteraceae family) essential oil constituents of the aerial parts harvested during the vegetative and flowering stages was greater than that of the fruit set stage, which was consistent with our finding that there were significantly more *A. ordosica* oil components identified during the vegetative stage (74 components). The difference may be related to the plant harvest period since the aerial parts of *A. ordosica* were harvested during the vegetative period instead of the flowering or fruit set stage in our study. The phenological stage significantly influences the yield, chemical composition, and biological properties of essential oils. Meanwhile, we found that terpenoids comprised over 50% of the entire *A. ordosica* oil across all growth stages. As major components, spathulenol, nerolidol, and D-limonene were identified at both the vegetative and flowering stages. In the present study, *α*-curcumene (9.24%, sesquiterpene) is reported for the first time as the predominant compound in the essential oil of *A. ordosica* during the vegetative period. These results indicate that the vegetative-stage essential oil of *A. ordosica* may have distinct biological functions from the flowering and fruit set-stage essential oils.

Prior studies indicated that *Artemisia* plants in the Asteraceae family had moderate antioxidant activity potential. *Artemisia scoparia* residue essential oil displayed high DPPH and OH^●^ radical scavenging activities, with IC_50_ values of 146.3 μL/mL and 145.2 μL/mL, respectively [11]. However, the scavenging ability of the *Artemisia chamaemelifolia* essential oil obtained during the vegetative and 50% flowering stages was weaker, with IC_50_ values ranging from 310.1 to 809.8 μL/mL [12]. Moreover, the flowering-stage essential oil of *Artemisia campestris* exhibited much weaker efficacy against the DPPH radical, with an IC_50_ value of 94.5 mg/mL [13]. Similarly, the present study demonstrated that the *A. ordosica* essential oil had modest antioxidant activity against DPPH (IC_50_ value of 5.9 mg/mL), ABTS (IC_50_ value of 5.7 mg/mL), and OH^●^ (IC_50_ value of 6.8 mg/mL) radicals. Lopes-Lutz et al. [14] found that essential oils of a number of *Artemisia* species possessed weak antioxidant properties to prevent the reduction of DPPH radicals. Generally, the antioxidant capacity of essential oils is linked to the presence of phenolic substances, oxygenated monoterpenes, or monoterpene hydrocarbon fractions [7,15,16]. During the plant growth season, a negative correlation has been found between monoterpene and sesquiterpene chemicals [17]. Consistent with previous findings, the ratios of phenolic compounds (0.49%) and monoterpene compounds (27.69%) were relatively lower in the essential oil of *A. ordosica* collected at the vegetative stage in the present study, whereas the ratio of sesquiterpene compounds was relatively greater at 51.81%. However, the antioxidant activity of the flowering and fruit set-stage essential oil of *A. ordosica* has not been investigated, and requires further research due to the variation in essential oil components and composition.

In the current study, the vegetative-stage essential oil of *A. ordosica* demonstrated remarkable bactericidal activity, particularly against *S. aureus* (Gram-positive), *E. coli* (Gram-negative), and *S. abony* (Gram-negative), which are significant pathogens that cause severe economic losses in animal production in agriculture. However, the influence of the *A. ordosica* essential oil on antibacterial activity has not previously been described. Researchers have attributed the antibacterial activity of plant essential oils to enriched terpenoids [18]. Sesquiterpenes, such as spathulenol, *α*-curcumene, *α*-bisabolol and nerolidol, were the main components of the *A. ordosica* essential oil in the present work. Their high hydrophobicity facilitates penetration across the plasma membrane and interaction with intracellular proteins and/or intraorganelle sites [19,20,21,22]. Extensive leakage from microbial cells or the outflow of essential molecules and ions (particularly K^+^ ions) may result in cell death [23]. In addition to causing membrane disruption, nerolidol may also be responsible for the downregulation of *α*-hemolysin gene *HLA* expression in *S. aureus*, as demonstrated by quantitative real-time PCR [24]. However, the potential synergistic effects of minor components on the antibacterial activity should also be considered [25]. According to previous studies, *α*-pinene, *β*-pinene and limonene are capable of degrading cellular integrity, increasing cell permeability, and inhibiting respiration and ion transport [26,27,28]. The fractions with appropriate MIC values of *α*-pinene, *β*-pinene and limonene can inhibit Gram-positive and Gram-negative bacteria considerably [21,26,27,28]. Therefore, the essential oil of *A. ordosica* may have multiple mechanisms for killing microbial cells. However, in vitro studies should also be conducted to observe the toxic concentration of these bioactive compounds, especially with the knowledge that sesquiterpenes can be hepatotoxic [29].

Furthermore, *S. aureus* was more susceptible to the *A. ordosica* essential oil than *E. coli* and *S. abony*, according to our experimental findings. The variable degree of sensitivity of the *A. ordosica* essential oil against these bacterial strains might be attributed to the difference in cell wall structure between Gram-positive and Gram-negative bacteria [28]. According to a previous study, Gram-negative bacteria, such as *E. coli* and *S. abony*, have a stiff outer lipopolysaccharide layer, inhibiting the diffusion of hydrophobic substances [30]. While Gram-positive bacteria have a monopeptide layer, it is not thick enough to resist small antibacterial compounds, allowing easy access to the cell membrane of Gram-positive bacteria [31]. Additional investigations have shown that the lipophilic ends of the lipoteichoic acid present in the cell membrane of Gram-positive bacteria may facilitate the entry of hydrophobic components of essential oils [32,33]. Thus, the identification of alternative compounds to control microbial strains is important for the management of damage caused by these pathogens due to stress in the animal gut, especially given the increase in resistance to conventional chemical products. Meanwhile, it should be noted that, despite the favorable in vitro potential of any compound or extract from plants, the safety level and the beneficial outcomes could only be determined through in vivo toxicological studies [34].

## 4. Materials and Methods

### 4.1. Plant Material and Essential Oil Extraction

Fresh aerial parts of wild *A. ordosica* (Figure 4) were collected in June 2020 from the study site on the eastern edge of the Kubuqi Desert (40°19′ N and 109°59′ E, Inner Mongolia). After identification, the voucher specimen (CAAS-G-20200610) was deposited at the Herbarium of the Chinese Academy of Agricultural Sciences, Grassland Research Institute (Hohhot, China). The aerial parts of plant samples were air-dried at room temperature and ground, and then, 25 g of each sample was individually subjected to hydrodistillation for essential oil extraction (3 h) using a Clevenger-type apparatus. After extraction, the oils were dehydrated with anhydrous sodium sulfate and stored in dark, tightly sealed vials at 4 °C until further examination. The essential oil yield was expressed in terms of dry weight.

### 4.2. Chemical Composition Analysis of the Essential Oil

The chemical compositions of the *A. ordosica* essential oil samples were characterized using a gas chromatography system coupled with mass spectrometry (GC-MS) (Agilent 19091J-433, Santa Clara, CA, USA) equipped with a capillary column (30 m × 250 μm × 0.25 μm) at the programmed temperature from 40 °C/2 min to 250 °C (3 °C/min) and maintained at this temperature for 20 min. Helium was used as the carrier gas at a constant flow rate of 1 mL/min. The injector temperature was 250 °C with an injection volume of 1 μL and a split ratio of 20:1. The transfer line, ion source, and quadrupole temperatures were 285, 230, and 150 °C, respectively. For mass spectrometry, a scan range of 40 to 550 *m/z* and a solvent delay of 3 min were used. The retention indices were calculated using a homologous series of *n*-alkanes (C_7_–C_28_) (Sigma-Aldrich, St. Louis, MO, USA). The components were identified through a comparison of the experimental mass spectra to those in the spectrometer database (NIST MS Library v. 2.0) and published works.

### 4.3. Antioxidant Activity Assay

DPPH, ABTS, and OH^●^ assay methods were used to evaluate the antioxidant activity of the *A. ordosica* essential oil. Ascorbic acid was used as a standard. All tests were conducted in triplicate.

#### 4.3.1. DPPH Assay

DPPH scavenging activity was analyzed according to a previous study [35] with slight modifications. DPPH solution (0.005%) was prepared by dissolving 5 mg of DPPH in 100 mL of 98% methanal. The DPPH solution (2 mL) was added to 2 mL of various concentrations of the *A. ordosica* essential oil samples. After incubation in darkness at room temperature for 30 min, the absorbance of the samples was measured at 517 nm. The antioxidant activity was calculated as follows: (%, scavenging of DPPH) = [(A_control_ − A_sample_)/A_control_] × 100. Antioxidant activity was represented as IC_50_, the concentration of the essential oil sample providing 50% elimination.

#### 4.3.2. ABTS Assay

Antioxidant activity was determined according to radical scavenging ability using the ABTS radical scavenging method [36]. The absorbance of the samples was measured at 734 nm. The antioxidant activity was calculated as follows: (%, scavenging of ABTS) = [(A_control_ − A_sample_)/A_control_] × 100. Antioxidant activity was represented as IC_50_, stated as the concentration of the essential oil sample providing 50% elimination.

#### 4.3.3. OH^●^ Scavenging Activity

OH^●^ scavenging activity was measured by the deoxyribose degradation method [32]. The absorbance of the samples was determined at 532 nm. The antioxidant activity was calculated as follows: (%, scavenging of hydroxyl radical) = [(A_control_ − A_sample_)/A_control_] × 100. Antioxidant activity was represented as IC_50_, stated as the concentration of the essential oil sample providing 50% elimination.

### 4.4. Evaluation of Antibacterial Activity

The antibacterial activity of the *A. ordosica* essential oil was evaluated using microorganism strains, including *Staphylococcus aureus* ATCC 6538 (Gram-positive bacteria), *Escherichia coli* ATCC 8739 (Gram-negative bacteria), and *Salmonella abony* NTCC 6017 (Gram-negative bacteria). The antibacterial assay was conducted by the broth microdilution method [37]. Colonies of the microbial strains were maintained on nutrient agar (Hopebio, Qingdao, China) at 37 °C in the incubator. Using a twofold dilution approach, emulsions were created with dimethyl sulfoxide (DMSO) for each essential oil sample to establish a concentration serial from 40 to 2.5 μL/mL in the final test. The MIC of each pathogen was determined using a test tube containing 3460 μL of nutrient broth (Hopebio, Qingdao, China). After adding the essential oil emulsion (500 μL) and broth, 40 μL of inoculum at a concentration of 3 × 10^9^ CFU/mL was added to each test tube. In the bacterial growth control group, neither essential oil nor detergent was applied to the test tubes. Each tube was visually checked for the existence of turbidity after 18 h of incubation at 37 °C. The MIC value is the concentration at which observable microorganism growth is inhibited relative to the control group. All tests were conducted in triplicate under aerobic conditions.

### 4.5. Statistics

All analyses and measurements were performed in triplicate. Data from each experiment were statistically analyzed using GraphPad Prism 9.0 software (GraphPad Software Inc., San Diego, CA, USA).

## 5. Conclusions

This is the first study on the essential oil composition and bioactivity of *A. ordosica* obtained during the vegetative period. The vegetative stage aerial parts of *A. ordosica* presented a decent essential oil yield (0.36%, *v*/*w*) with 74 components, and the major components were spathulenol and *α*-curcumene, followed by *α*-bisabolol, *cis*-(+)-nerolidol and D-limonene. The essential oil of *A. ordosica* showed modest antioxidant activity, although its radical scavenging activity was lower than that of the control and other *Artemisia* species in the Asteraceae family. However, the generated oils exhibited significant antibacterial activity against highly susceptible strains of pathogenic and spoilage bacteria, such as *S. aureus*, *E. coli*, and *S. abony*. The *A. ordosica* essential oil may be a viable alternative to antibiotics in animal nutrition applications for a more sustainable feed industry. Considering the safety level and the beneficial outcomes of the potential utilization of the *A. ordosica* essential oil, in vivo toxicological studies are still required in future research.

## Figures and Tables

**Figure 1 molecules-27-08898-f001:**
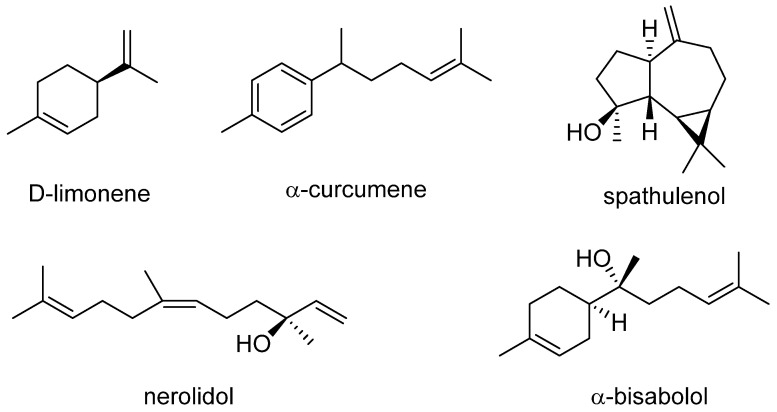
Major chemical constituents identified in the essential oil of *A. ordosica*.

**Figure 2 molecules-27-08898-f002:**
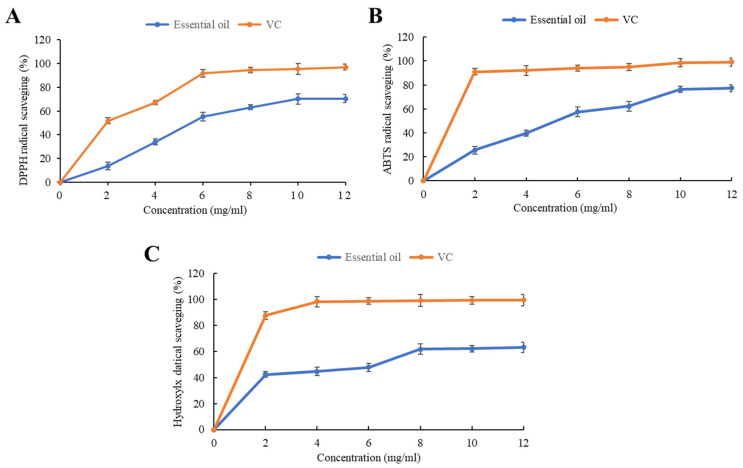
Antioxidant activities of the vegetative-stage essential oil of *A. ordosica*. (**A**) DPPH free radical scavenging activity. (**B**) ABTS free radical scavenging activity. (**C**) Hydroxyl radical scavenging activity. All the results are compared with a standard (ascorbic acid) in a line chart.

**Figure 3 molecules-27-08898-f003:**
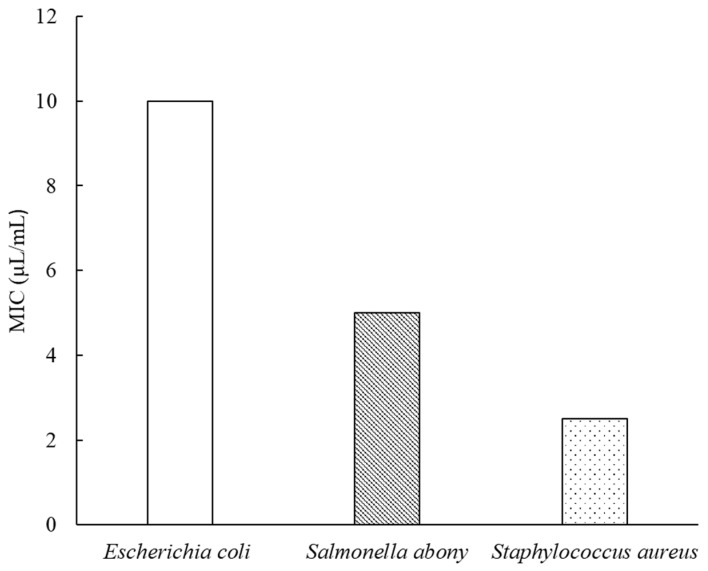
Minimum inhibitory concentrations (MICs) of *A. ordosica* essential oil against tested bacterial strains.

**Figure 4 molecules-27-08898-f004:**
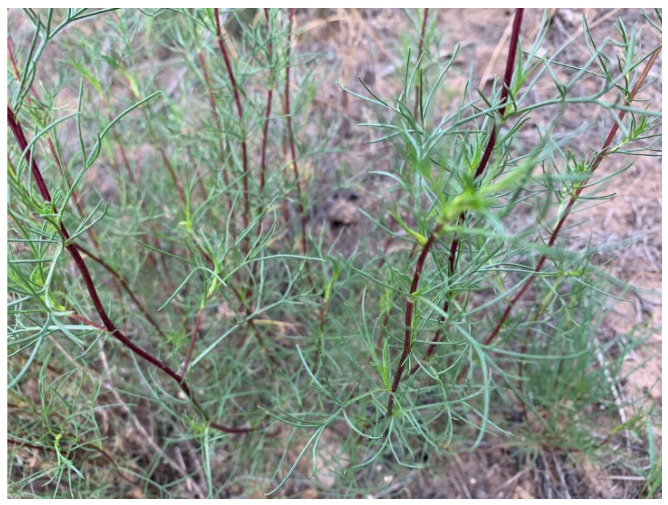
Photography of aerial parts of wild *Artemisia ordosica* collected at the vegetative stage.

**Table 1 molecules-27-08898-t001:** Chemical composition of the essential oil extracted from the aerial parts of *Artemisia ordosica* through hydrodistillation.

No.	Compounds	Area (%)	RI
1	4-Carene	0.53	919
2	Sulcatone	0.12	938
3	*β*-Pinene	2.30	943
4	*α*-Pinene	2.34	948
5	*β*-Phellandrene	1.76	964
6	Cosmene	0.41	966
7	Epoxycyclooctane	0.13	970
8	(E)-*β*-Ocimene	1.35	976
9	*β*-Ocimene	1.06	976
10	2,5-Etheno[4.2.2]propella-3,7,9-triene	0.98	976
11	Benzaldehyde	1.69	982
12	1-Acetyl-2-methyl-1-cyclopentene	0.35	996
13	*γ*-Terpinene	1.63	998
14	d-Limonene	4.25	1018
15	1,3,8-*p*-Menthatriene	0.26	1029
16	δ-Terpinolene	0.63	1052
17	2-Methyl-butyric acid 2-methylbutyl ester	0.30	1054
18	Cineol	0.30	1059
19	(±)-Cryptone	0.67	1069
20	(±)-Linalool	0.53	1082
21	Pinocarvone	0.32	1114
22	Isovaleric acid pentyl ester	0.38	1118
23	*α*-Phellandren-8-ol	3.12	1125
24	3-Caren-2-ol	0.91	1131
25	Verbenol	0.76	1136
26	(-)-4-Terpineol	1.24	1137
27	*cis*-Piperitol	0.13	1175
28	1,6-Methanocyclodecapentaene	0.08	1189
29	*trans*-3-Hexenyl butanoate	0.37	1191
30	*p*-Cymene-8-ol	0.70	1197
31	1-Phenylpenta-2,4-diyne	0.26	1206
32	Hexyl isovalerate	0.89	1218
33	Leaf 2-methylbutyrate	0.41	1226
34	Geraniol	0.76	1228
35	Perillol	0.33	1261
36	Analgit	0.50	1281
37	*cis*-3-Hexenyl pentanoate	1.21	1290
38	*trans*-2-Hexenyl valerate	0.31	1290
39	4-Vinylguaiacol	0.49	1293
40	3a,4,5,6,7,7a-Hexahydro-4,4,7a-trimethyl-1H-inden-1-one	0.15	1343
41	Alloaromadendrene	0.87	1386
42	Berkheyaradulene	1.32	1416
43	Dihydropseudoionone	0.52	1420
44	*α*-Bergamotene	1.32	1430
45	Elixene	0.04	1431
46	1-Epibicyclosesquiphellandrene	0.31	1435
47	Zingiberene	1.25	1451
48	*β*-Phenylethyl butyrate	0.17	1458
49	Capillin	0.89	1461
50	4-Methylene-1-methyl-2-(2-methyl-1-propen-1-yl)-1-vinyl-cycloheptane	0.04	1475
51	1-(1,5-Dimethyl-5-hexenyl)-4-methyl-1,4-cyclohexadiene	2.33	1480
52	4-(2,4,4-Trimethylbicyclo[4.1.0]hept-2-en-3-yl)-(*E*)-3-buten-2-one	0.73	1480
53	*γ*-Undecalactone	0.33	1483
54	*α*-Bulnesene	1.14	1490
55	Citronellyl butyrate	1.14	1501
56	*α*-Curcumene	9.24	1524
57	Spathulenol	9.93	1536
58	Geranyl butyrate	3.04	1550
59	Nerolidol	5.26	1564
60	Nerolidol B	2.34	1564
61	11-Tridecyn-1-ol	0.43	1574
62	Neryl 2-methyl butyrate	1.32	1586
63	*β*-Bisabolol	1.08	1619
64	*α*-Bisabolol	6.45	1625
65	*δ*-Terpineol pentanoic ester	0.73	1626
66	2-Hydroxy-eudesmane-4,11-diene	0.34	1690
67	(-)-*α*-Bisabolol oxide B	0.73	1707
68	Farnesol	0.18	1710
69	1,5,5,8-Tetramethyl-3,7-cycloundecadien-1-ol	0.55	1719
70	Nerolidyl acetate	3.76	1754
71	4-Methoxystyrene	0.19	1754
72	3-Hydroxy-humulane-1,6-dien	1.08	1757
73	*δ*-Cuparenol	0.38	1776
74	Bisaboloxide A	0.28	1798
	Monoterpenes hydrocarbons	16.51	
	Oxygenated monoterpenes	11.59	
	Sesquiterpene hydrocarbons	17.86	
	Oxygenated sesquiterpenes	33.95	
	Phenolic compounds	0.49	
	Esters	9.03	
	Others	5.10	
	Total	94.53	

**Table 2 molecules-27-08898-t002:** Inhibitory effects of the *Artemisia ordosica* essential oil on the three microbial strains.

Name of Micro-Organism	Concentration of Oil (μL/mL)					
	0	2.5	5	10	20	40
*Staphylococcus aureus* (ATCC 6538)	+	−	−	−	−	−
*Escherichia coli* (ATCC 8739)	+	+	+	−	−	−
*Salmonella abony* (NTCC 6017)	+	+	−	−	−	−

Note: −, no visible growth of microorganisms; +, evidently visible growth of microorganisms.

## Data Availability

Not applicable.

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
