# Peer review of "Chemical Composition and Antioxidant and Antibacterial Potencies of the Artemisia ordosica Aerial Parts Essential Oil during the Vegetative Period"

_molecules, 2022, doi:10.3390/molecules27248898_

Round 1

Reviewer 1 Report

The manuscript entitled “Chemical composition, antioxidant and antimicrobial potencies of essential oil of Artemisia ordosica aerial parts at vegetative period” presents a significant work. In my opinion, the presented research is well prepared and will be interesting to readers. Therefore, I recommend the manuscript to be published in Molecules after minor revision. My comments are only related to some typos and technical mistakes.

1.     Escherichia coli should be used in the abstract instead of Escherichia colis, same on the line 97

2.     On line 31, please add examples of the best known Artemisia species with medicinal properties

3.     In line 49 there are no references to the described studies

4.     In line 108, 112, 122 please write spathulenol in lowercase

5.     In line 111 please write 2,5-etheno[4.2.2]propella-3,7,9-triene in lowercase

6.     In line 112, 123, 164, 166 please write D-limonene in lowercase

7.     Were the DPPH tests based on the previously described method? No references

8.     Please explain the abbreviation MIC when it first appears in the text

9.     Please standardize the writing of compounds with lowercase or uppercase letters in Table 1

 Please read the discussion carefully and correct it, as there are a lot of typos in it. Especially check the names of the chemical compounds

Author Response

The manuscript entitled “Chemical composition, antioxidant and antimicrobial potencies of essential oil of Artemisia ordosica aerial parts at vegetative period” presents a significant work. In my opinion, the presented research is well prepared and will be interesting to readers. Therefore, I recommend the manuscript to be published in Molecules after minor revision. My comments are only related to some typos and technical mistakes.

  1. Escherichia coli should be used in the abstract instead of Escherichia colis, same on the line 97

Response: Revised. See line 21and 114.

  1. On line 31, please add examples of the best known Artemisia species with medicinal properties

Response: Revised. We’ve added Artemisia species with medicinal properties. See line 32-34.

  1. In line 49 there are no references to the described studies

Response: We’ve changed the description of the sentence. See line 51-55.

  1. In line 108, 112, 122 please write spathulenol in lowercase

Response: Revised. See line 125, 129 and 139.

  1. In line 111 please write 2,5-etheno[4.2.2]propella-3,7,9-triene in lowercase

Response: Revised. See line 128.

  1. In line 112, 123, 164, 166 please write D-limonene in lowercase

Response: Revised. See line 129, 140, 181 and 183.

  1. Were the DPPH tests based on the previously described method? No references

Response: Revised. We’ve added the reference used for measuring DPPH free radical scavenging activity. See reference No. 35 in revised manuscript.

  1. Please explain the abbreviation MIC when it first appears in the text

Response: Revised. We’ve added the whole name of MIC when it first appears in the Abstract and the text. See line 21 and 106.

  1. Please standardize the writing of compounds with lowercase or uppercase letters in Table 1

Response: Revised. We’ve rechecked the name of compounds and the writing of Table.

Please read the discussion carefully and correct it, as there are a lot of typos in it. Especially check the names of the chemical compounds

Response: Revised. We’ve rechecked the discussion part and corrected writing errors.

Reviewer 2 Report

This manuscript reviews 74 chemical components isolated from Artemisia ordosica essential oils by gas chromatography coupled with mass spectrometry (GC-MS) analysis and it has been tested for antioxidant against 2,2-Diphenylhydrazyl (DPPH), 2,2’-Azino-bis (3-ethylbenzothiazoline-6- sulfonic acid (ABTS), and hydroxyl radical (OH) radicals and antimicrobial against Staphylococcus aureus, Salmonella abony and Escherichia colis activities, which has shown promising results.

But there is doubt that the scientific novelty of the study and its relevance lies in the fact that the study on the essential oil composition and bioactivity of A. ordosica was conducted for the first time?

In section 4 ‘Materials and Methods. Plant Material and Essential Oil Extraction’, it is given ‘aerial parts’ and ‘fresh plant samples’. Does it mean that this plant has been divided into several parts, for example stems, leaves, petioles, flowers, fruit, seeds and has been extracted according to the property of each part? If yes, then it is not specified which parts?

Recently, Jia-Wei Zhang et al. (2022, https://doi.org/10.3390/plants11131627) have been studied and published essential oils and biological activities of five Artemisia species in which including Artemisia ordosica. The author compared and included the results of the study in your manuscript?

Add future perspective of this study. Also add potential of the studies species as industrial important plant and facilitating pharmaceutical (or other) industries in specific areas.

The scientific novelty, data, and results of the manuscript as well as complete literature review on related studies are lacking.

The manuscript results are not enough for publication in Molecules, and the manuscript is less than average level manuscript from Molecules.

Based on what is mentioned above, I recommended the rejection of the manuscript. Maybe it could be corrected or more scientific results to be added.

Author Response

This manuscript reviews 74 chemical components isolated from Artemisia ordosica essential oils by gas chromatography coupled with mass spectrometry (GC-MS) analysis and it has been tested for antioxidant against 2,2-Diphenylhydrazyl (DPPH), 2,2’-Azino-bis (3-ethylbenzothiazoline-6- sulfonic acid (ABTS), and hydroxyl radical (OH●) radicals and antimicrobial against Staphylococcus aureus, Salmonella abony and Escherichia coli activities, which has shown promising results.

But there is doubt that the scientific novelty of the study and its relevance lies in the fact that the study on the essential oil composition and bioactivity of A. ordosica was conducted for the first time?

Response: We’ve mentioned in the first paragraph of the revised manuscript that the essential oil of A. ordosica collected during flowering and fruit set stages focused on the allelochemicals and larvicidal effects of A. ordosica. While a number of research demonstrated that plants collected during the vegetative period had the optimum antioxidant or antibacterial activities, which had not been studied on A. ordosica essential oil collected during the vegetative period. Furthermore, monitoring the phytochemical content and quality during phenological growth stages can indicate the pattern of accumulation of essential oil compounds for acquiring specific phytochemical components with desired quality for pharmaceutical or animal feed industries. Therefore, the obtained result of our study is great complement for understanding the composition changes in A. ordosica essential oil during the entire phenological growth stages, which is beneficial for the utilization of A. ordosica essential oil in multiple fields in the future. We’ve also revised the corresponding text in the “Introduction” section in the manuscript.

In section 4 ‘Materials and Methods. Plant Material and Essential Oil Extraction’, it is given ‘aerial parts’ and ‘fresh plant samples’. Does it mean that this plant has been divided into several parts, for example stems, leaves, petioles, flowers, fruit, seeds and has been extracted according to the property of each part? If yes, then it is not specified which parts?

Response: Revised. We’ve revised the description of plant samples, which may cause misleading of the reader. See line 211.

Recently, Jia-Wei Zhang et al. (2022, https://doi.org/10.3390/plants11131627) have been studied and published essential oils and biological activities of five Artemisia species in which including Artemisia ordosica. The author compared and included the results of the study in your manuscript?

Response: Jia-Wei Zhang et al. (2022, https://doi.org/10.3390/plants11131627) clearly mentioned in the “Result section” of article that the chemical composition data of A. ordosica essential oil were drawn from Zhang et al. (2017, http://dx.doi.org/10.1016/j.indcrop.2017.02.020). We’ve cited the above article and compared the results with ours (see reference No. 4 and related text in the manuscript). Moreover, Jia-Wei Zhang et al. (2022) focused on the anti-insect activity of A. ordosica essential oil, which is not related to the antioxidant and antibacterial activities that discussed in our manuscript.

Add future perspective of this study. Also add potential of the studies species as industrial important plant and facilitating pharmaceutical (or other) industries in specific areas.

Response: We’ve added future perspective and potential of our study. See revised Abstract, Introduction, Discussion, and Conclusion sections.

The scientific novelty, data, and results of the manuscript as well as complete literature review on related studies are lacking.

The manuscript results are not enough for publication in Molecules, and the manuscript is less than average level manuscript from Molecules.

Based on what is mentioned above, I recommended the rejection of the manuscript. Maybe it could be corrected or more scientific results to be added.

Response: We’ve mentioned in the first paragraph of the revised manuscript that the essential oil of A. ordosica collected during flowering and fruit set stages focused on the allelochemicals and larvicidal effects of A. ordosica. While a number of research demonstrated that plants collected during the vegetative period had the optimum antioxidant or antibacterial activities, which had not been studied on A. ordosica essential oil collected during the vegetative period. The present study indicated that A. ordosica essential oil collected during the vegetative period had great potential to be alternative to antibiotics in animal feed industry. Thus, our results obtained in this study are great complements to the development of A. ordosica essential oil industry in the future.

Reviewer 3 Report

Reviewer comments for authors:

The manuscript entitled “Chemical Composition, Antioxidant and Antimicrobial Potencies of Essential Oil of Artemisia ordosica Aerial Parts at Vegetative Period” are reviewed. The presented manuscript abounds with serious problems that require its major revision

1.      In Table 1, include the biomedical applications with their references

2.      Essential Oil of Artemisia ordosica and their chemical composition, biomedical applications were already studied. Many articles were available. What is a novelty in this work?

3.      Include the chromatogram for GC-MS analysis

4.      Include, antibacterial activity plate images

5.      Include the confirmation of MIC done by Resozurin dye assay

6.      Change antimicrobial activity to antibacterial activity through the manuscript

7.      What is the criteria for choosing the organisms for antibacterial activity?

8.      Correct the grammatical errors throughout the manuscript. The manuscript needs to be completely and thoroughly revised for both English and scientific style by a professional scientific reviewer. The meaning and consists of each sentence should be double-checked.

Author Response

The manuscript entitled “Chemical Composition, Antioxidant and Antimicrobial Potencies of Essential Oil of Artemisia ordosica Aerial Parts at Vegetative Period” are reviewed. The presented manuscript abounds with serious problems that require its major revision.

  1. In Table 1, include the biomedical applications with their references

Response: A number of compounds identified in this study still lack biomedical applications, which require further research in the future. The biomedical applications of major constituents of our study were already include in the manuscript. Previous published short communications related to plant essential oil in Molecules also did not include component biomedical applications. So, our manuscript is short communication for brief information of the composition and some important in vitro bioactivities of the vegetative-stage essential oil of A. ordosica. Thus, the content of manuscript is still limited compared to full research articles or comprehensive reviews.

Article list:

de Moraes, Â.A., de Jesus Pereira Franco, C., Ferreira, O.O., Varela, E.L., do Nascimento, L.D., Cascaes, M.M., da Silva, D.R., Percário, S., de Oliveira, M.S., & de Aguiar Andrade, E.H. (2022). Myrcia paivae O.Berg (Myrtaceae) essential oil, first Study of the chemical composition and antioxidant potential. Molecules, 27.

Fidan, H., Stefanova, G., Kostova, I., Stankov, S., Damyanova, S., Stoyanova, A., & Zheljazkov, V.D. (2019). Chemical composition and antimicrobial activity of laurus nobilis L. essential oils from Bulgaria. Molecules, 24.

  1. Essential Oil of Artemisia ordosica and their chemical composition, biomedical applications were already studied. Many articles were available. What is a novelty in this work?

Response: Previous research related to the essential oil of A. ordosica collected during flowering and fruit set stages focused on the allelochemicals and larvicidal effects of A. ordosica. While a number of research demonstrated that plants collected during the vegetative period had the optimum antioxidant or antibacterial activities, which had not been studied on A. ordosica essential oil collected during the vegetative period. Monitoring the phytochemical content and quality during phenological growth stages can indicate the pattern of accumulation of essential oil compounds for acquiring specific phytochemical components with desired quality for pharmaceutical or food industries. Therefore, the obtained result of our study is great complement for understanding the composition changes in A. ordosica essential oil during the entire phenological growth stages, which is beneficial for the utilization of A. ordosica essential oil in multiple fields in the future. We’ve also revised the corresponding text in the “Introduction” section in the manuscript.

  1. Include the chromatogram for GC-MS analysis

Response: As suggested by reviewer, we’ve added the chromatogram for GC-MS analysis. See supplemented file: Figure S1.

  1. Include, antibacterial activity plate images

Response: Plate method and broth microdilution test are both acceptable for evaluation of antibacterial activity. We’ve tried the plate method, but it was not ideal for evaluating A. ordosica essential oil due to its relatively low diffusivity in nutrient agar. Thus, we used the broth microdilution test, which clearly showed the antibacterial activity of A. ordosica essential oil.

  1. Include the confirmation of MIC done by Resozurin dye assay

Response: The suggestion provided is very beneficial for the manuscript. However, we’ve noticed that the MIC value obtained by broth microdilution test followed visual turbidity confirmation is also acceptable in following peer-reviewed articles. We hope the answer can be satisfied by the reviewer.

Article list:

Falci, S.P., Teixeira, M.A., Chagas, P.F., Martínez, B.B., Loyola, A.B., Ferreira, L.M., & Veiga, D.F. (2015). Antimicrobial activity of Melaleuca sp. oil against clinical isolates of antibiotics resistant Staphylococcus aureus. Acta cirurgica brasileira, 30 7, 491-6.

Ács, K., Balázs, V.L., Kocsis, B., Bencsik, T., Böszörményi, A., & Horváth, G. (2018). Antibacterial activity evaluation of selected essential oils in liquid and vapor phase on respiratory tract pathogens. BMC Complementary and Alternative Medicine, 18.

  1. Change antimicrobial activity to antibacterial activity through the manuscript

Response: As suggested by the reviewer, we’ve changed the words in the revised manuscript.

  1. What is the criteria for choosing the organisms for antibacterial activity?

Response: Staphylococcus aureus (Gram-positive bacteria), Escherichia coli (Gram-negative bacteria), and Salmonella abony (Gram-negative bacteria) are typical bacteria strains for evaluating antibacterial activity of plant essential oils, which induced great damages in agriculture as we mentioned in the manuscript. A number of studies have used these strains for testing. See following article list using these bacterial strains for evaluating antibacterial activity of plant essential oils:

Cazella, L.N., Glamočlija, J., Soković, M.D., Gonçalves, J.E., Linde, G.A., Colauto, N.B., & Gazim, Z.C. (2019). Antimicrobial activity of essential oil of Baccharis dracunculifolia DC (Asteraceae) aerial parts at flowering period. Frontiers in Plant Science, 10.

Fidan, H., Stefanova, G., Kostova, I., Stankov, S., Damyanova, S., Stoyanova, A., & Zheljazkov, V.D. (2019). Chemical composition and antimicrobial activity of Laurus nobilis L. essential oils from Bulgaria. Molecules, 24.

  1. Correct the grammatical errors throughout the manuscript. The manuscript needs to be completely and thoroughly revised for both English and scientific style by a professional scientific reviewer. The meaning and consists of each sentence should be double-checked.

Response: Revised. The revision of the manuscript was accomplished by American Journal Experts (AJE) with certification uploaded in the supplementary material.

Reviewer 4 Report

The manuscript examined the chemical composition and evaluated some biological activities of the essential oil extracted from the aerial parts of a less studied plant, Artemisia ordosicaI would recommend this manuscript for publication after the next suggestions have been attended to:

- if not your own developed assay, all methods should have a reference (see DPPH)

- Lines 61-64: can you introduce a figure with the chemical structure of these 5 compounds?

Line 97: E. coli not colis

- please introduce in Discussion the following important ideas and adjust accordingly the last sentence in Abstract and Conclusions

1) in vitro studies should be conducted to observe the toxic concentration of these bioactive compounds, with the knowledge that sesquiterpenes can be hepatotoxic (He et al. Herb-Induced Liver Injury: Phylogenetic Relationship, Structure-Toxicity Relationship, and Herb-Ingredient Network Analysis. Int J Mol Sci. 2019;20(15):3633. doi: 10.3390/ijms20153633)

2) despite the favorable in vitro potential of any compound or extract, the safety level and the beneficial outcomes could only be determined through in vivo toxicological studies (Vedeanu et al. Subacute co-exposure to low doses of ruthenium(III) changes the distribution, excretion and biological effects of silver ions in rats. Environ Chem. 2019;17:163-172. doi: 10.1071/EN19249)

Author Response

The manuscript examined the chemical composition and evaluated some biological activities of the essential oil extracted from the aerial parts of a less studied plant, Artemisia ordosica. I would recommend this manuscript for publication after the next suggestions have been attended to:

1- if not your own developed assay, all methods should have a reference (see DPPH)

Response: Revised. We’ve added the reference used for measuring DPPH free radical scavenging activity. See reference No. 35 in revised manuscript.

2- Lines 61-64: can you introduce a figure with the chemical structure of these 5 compounds?

Response: Revised. We’ve added a figure with the chemical structure of these 5 compounds. See Figure 1 in the revised manuscript.

3- Line 97: E. coli not colis

Response: Revised. See line 114.

4- please introduce in Discussion the following important ideas and adjust accordingly the last sentence in Abstract and Conclusions

1) in vitro studies should be conducted to observe the toxic concentration of these bioactive compounds, with the knowledge that sesquiterpenes can be hepatotoxic (He et al. Herb-Induced Liver Injury: Phylogenetic Relationship, Structure-Toxicity Relationship, and Herb-Ingredient Network Analysis. Int J Mol Sci. 2019;20(15):3633. doi: 10.3390/ijms20153633)

2) despite the favorable in vitro potential of any compound or extract, the safety level and the beneficial outcomes could only be determined through in vivo toxicological studies (Vedeanu et al. Subacute co-exposure to low doses of ruthenium (III) changes the distribution, excretion and biological effects of silver ions in rats. Environ Chem. 2019;17:163-172. doi: 10.1071/EN19249)

Response: The suggested provided is very helpful for increasing the quality of our manuscript. We’ve added the above information in the revised manuscript. See line 24-26, 185-187, and 201-204.

Round 2

Reviewer 2 Report

The present study aimed to characterize the content and constituents of essential oil from aerial parts of A. ordosica collected during the vegetative stage, as well as to determine its free radical scavenging and antimicrobial properties. The methods for evaluating antioxidant capacity are primitive and must be proven by more complex evaluation systems with more accurate equipment such as ESR than simple reactions with radicals.
As authors mentioned there is already present few reports about essential oil constituents, which casts doubt on the novelty and significance of your work. They also discussed about the other results in similar species.
I suggest evaluating activity in cell or animal base system, there is many kinds of experiments where antioxidants can be applicable.

Reviewer 3 Report

From pathogenic bacterial strains were obtained from which place? MTCC or ATCC?

Include new plant image (Figure 4)